# A Prospective Study of Association of Micronutrients Deficiencies during Pregnancy and Neonatal Outcome among Women after Bariatric Surgery

**DOI:** 10.3390/jcm10020204

**Published:** 2021-01-08

**Authors:** Guillaume Ducarme, Lucie Planche, Emeric Abet, Valérie Desroys du Roure, Amélie Ducet-Boiffard

**Affiliations:** 1Department of Obstetrics and Gynecology, Centre Hospitalier Departemental, 85000 La Roche sur Yon, France; 2Clinical Research Center, Centre Hospitalier Departemental, 85000 La Roche sur Yon, France; lucie.planche@chd-vendee.fr (L.P.); valerie.desroysduroure@chd-vendee.fr (V.D.d.R.); 3General Surgery Department, Bariatric Surgery Program, Centre Hospitalier Departemental, 85000 La Roche sur Yon, France; emeric.abet@chd-vendee.fr; 4Department of Biology, Centre Hospitalier Departemental, 85000 La Roche sur Yon, France; 5Department of Endocrinology, Bariatric Surgery Program, Centre Hospitalier Departemental, 85000 La Roche sur Yon, France; amelie.ducet-boiffard@chd-vendee.fr

**Keywords:** bariatric surgery, pregnancy, micronutrients, selenium, birth weight, neonatal morbidity

## Abstract

Little is known about the association of micronutrients deficiencies during pregnancy and neonatal outcome among women after Roux-en-Y gastric bypass (RYGB) or sleeve gastrectomy (SG). We present a prospective study of 87 consecutive women with a history of RYGB (*n* = 37) or SG (*n* = 50) who underwent complete and regular clinical and biological nutritional assessments during pregnancy. Data on maternal characteristics, biological nutritional parameters, antenatal management, and perinatal outcome were collected. The objective was to evaluate serum levels of micronutrients at the second trimester, and to assess the association between micronutrients deficiencies and neonatal outcome. After RYGB, maternal age (34.3 ± 4.9 vs. 31.1 ± 4.6 years; *p* = 0.003) and pregnancy timing after surgery (64.5 ± 47.0 vs. 38.1 ± 28.1 months; *p* = 0.002) were significantly higher, and pre-pregnancy BMI was significantly lower (29.8 ± 5.6 vs. 32.9 ± 7.1 kg/m^2^; *p* = 0.03). Maternal and neonatal outcomes were similar. Additional supplementations after blood tests were similar in groups. Zinc, selenium, vitamins A1, B1, B6, C, and E levels were not different depending on the type of surgery. Zinc deficiency (7/87) was significantly more frequent after RYGP (18.9% vs. 0%; *p* = 0.02) and selenium deficiency (15/87) was similar in groups (21.6% after RYGB and 14.0% after SG; *p* = 0.36). The mean birth weight was significantly lower in selenium deficiency cases (3137 ± 550 vs. 3535 ± 737 g; *p* = 0.04). Selenium deficiency was negatively correlated with birth weight (*r* = −0.23; *p* = 0.03) and with birth weight z-score (*r* = −0.26; *p* = 0.01), but not correlated after adjustment for the procedure. The levels of micronutrients surveillance along pregnancy in women with a history of bariatric surgery is necessary to decrease the risk of inadequate fetal growth in the patients.

## 1. Introduction

The prevalence of obesity (body mass index (BMI) ≥ 30 kg/m^2^) in women in the reproductive age is a major public health problem with an increasing rate within the last 10 years (13.9% in 2009 to 15.7% in 2012 in France) [1]. During pregnancy, obesity clearly increases rates of obstetric complications (pre-eclampsia, gestational diabetes mellitus (GDM), and macrosomia) [2,3].

Bariatric procedures have a strong evidence of efficacy and safety [4]. Women in the reproductive age with a pregnancy desire are seeking bariatric surgery (BS), which could increase the pregnancy rate and improve pregnancy outcomes after weight loss by reducing the rates of GDM, pregnancy-induced hypertension, and macrosomia [5,6,7]. However, pregnant women after BS need to be evaluated carefully with increased risks of small-for-gestational age (SGA) fetus and prematurity [5,6,7]. Recent data have also suggested more frequent rates of long-term pediatric endocrine morbidity (hypothyroidism, diabetes mellitus, hypoglycemia or obesity), compared to obese and normal weight pregnant women without previous BS [8]. Furthermore, a recent systematic review found an unbalanced nutritional diet in patients after BS with inadequate protein intake and micronutrients deficiencies in many included studies [9]. As a consequence, current guidelines recommend routine testing and dietary supplementation before and after BS [10], especially for childbearing female and pregnant women [11]. However, there is a relative paucity of data concerning the obstetric and neonatal issues of micronutrients deficiencies during pregnancy after BS [12,13,14,15], and specifically after laparoscopic sleeve gastrectomy (SG), which should induce fewer nutritional deficiencies and fewer SGA, compared to Roux-en-Y gastric bypass (RYGB). The purpose of this study was to evaluate serum levels of micronutrients (zinc, selenium, vitamins A1, B1, B6, C, and E) during pregnancy in women after RYGB or SG, and thus to assess the association between nutritional deficiencies, and specifically selenium deficiency, and neonatal outcome.

## 2. Methods

### 2.1. Patient Selection

A retrospective review of prospectively collected data on consecutive cases of women with a history of BS who were admitted in an ambulatory procedure during pregnancy between January 2013 and July 2020 in a tertiary care hospital with more than 2600 annual deliveries was done. For this study, we included consecutive pregnant women with a history of RYGB or SG with complete clinical and biological nutritional assessments performed in our institution. We excluded women with a history of others types of BS (i.e., laparoscopic gastric banding, duodenal switch), women without gestational age dating (crown-to-rump length at a first-trimester ultrasound examination or fetal biometry before 24 weeks), multiple pregnancy, women with medical-indicated second trimester termination of pregnancy, and with intra uterine death and fetal loss before 22 weeks.

This present study was conducted in accordance with the French approved guidelines. All the participants received oral information on the study. Written consent was not required for the retrospective study according to the French law, but each woman got the opportunity to opt out of the analysis.

### 2.2. Management of Pregnant Women with Previous Bariatric Surgery

Surgical procedure and postoperative multidisciplinary management, especially after RYGB, were applied, as described elsewhere [16]. In our center, all pregnant women with a history of BS have a consultation before conception, or as soon as pregnancy is diagnosed, according to recommendations on bariatric surgery [11].

During pregnancy, they were managed monthly or more frequently if necessary, in an ambulatory obstetric unit with an obstetrician, physician with expertise in the nutritional management of BS patients, endocrinologist, registered dietitian to assess energy and protein intake with education, lifestyle intervention, and medical nutrition therapy.

Caloric allotment was based on the ideal body weight and weight gain for all women with a diet ranging from 25 kcal/kg for overweight and obese women to 35 kcal/kg for women with normal weight and divided into three full meals and four snacks of 50% complex carbohydrates, 30% fat, and 20% protein [17]. The recommended daily protein intake should be at least 60 g [11].

During pregnancy, all women were advised to take a daily multivitamin-mineral supplementary (one capsule daily) and folic acid up to 12 weeks, regardless of the type of BS. The multivitamin-mineral supplement recommended to the patients (Alvityl^®^, Urgo Healthcare, Chenove, France) contained 50 µg selenium and 10 mg of zinc per capsule alongside other vitamins and minerals. During each follow-up consultation, the women were encouraged to take their supplements daily. All women were scheduled for fasting venous blood tests at the 6th month of pregnancy. The prescription of other complements (including iron, vitamin D, vitamin B12, and calcium) was based on the patient’s information and on nutritional assays to maintain the levels of micronutrients as near to normal as possible [11]. The supplementations efficacy was determined by successive fasting venous blood tests during pregnancy [11].

GDM was diagnosed as usual, according to the international guidelines for pregnant women [18]: A fasting plasma glucose (FPG) at the first trimester of the pregnancy for women who belong to a risk group (e.g., age ≥ 35 years-old, BMI ≥ 25 kg/m^2^, familial history of diabetes, previous GDM or macrosomia), and a 75 g-oral glucose tolerance test (OGTT) between 24–28 weeks of gestation for pregnant women who presented risk factors of GDM and normal FPG at the first trimester. In pregnant women with a history of BS, 75 g-OGTT may not be well tolerated, especially after RYGB. Therefore, our multidisciplinary team has decided that a 1-week self-monitoring of capillary blood glucose with samples before each meal, and alternately 1 or 2 h after the start of each meal was used in all women after RYGB as a screening test after 24 weeks [19]. Once diagnosed, GDM was immediately managed by lifestyle modifications, self-monitoring of blood glucose, and insulin therapy if glycemic targets are not achieved [18].

All the included women had a fetal ultrasound assessment at 22, 28, 32, and 36 weeks of gestation to screen for small for gestational age fetus (SGA), defined as an ultrasonographic estimated fetal weight < 10th centile for gestational age on Hadlock curves [20] or large for gestational age fetus (LGA), defined as an estimated fetal weight > 90th centile for gestational age on Hadlock curves [20]. In the case of ultrasound suspicion of SGA or LGA, the evaluation and surveillance were adapted as per specific guidelines [21]. Women who presented SGA or LGA fetus at 37 weeks were advised to undergo induction of labor at 39 weeks of gestation. Women who presented normal fetal growth at the ultrasound assessment were allowed to await spontaneous labor until 41 weeks. There were no variations in practice as the same team cared for the cohort throughout the period of study.

### 2.3. Maternal and Perinatal Outcome

Maternal sociodemographic characteristics, information regarding pregnancy follow-up, and standard perinatal outcomes were collected prospectively by one obstetrician (GD), from a prospectively maintained database of women who were included in the study.

Maternal characteristics (age, geographic origin, type of BS, pre-pregnancy BMI, parity, pre-existing type 1 or 2 diabetes, medical history of chronic hypertension, and smoking during pregnancy) and antepartum variables (time to conception from bariatric surgery, GDM, pregnancy-induced hypertension/pre-eclampsia, intrahepatic cholestasis of pregnancy, nutritional supplementations after blood tests, gestational weight gain (GWG), and specific complications in pregnancy due to BS) were collected. Pre-pregnancy BMI was calculated (kilograms per square meter) from the database with the use of height and weight before the beginning of the pregnancy. As defined by the World Health Organization, women with a BMI of ≥30 kg/m^2^ were considered obese. Maternal gestational weight gain (GWG) was calculated as measured weight at the end of pregnancy minus the pre-pregnancy weight. Adequate weight gain during pregnancy was based on the Institute of Medicine’s GWG by maternal pre-pregnancy BMI recommendations [22]. Intrahepatic cholestasis of pregnancy is a cholestatic disorder characterized by pruritus with the onset in the second or third trimester of pregnancy and elevated serum aminotransferases and bile acid levels [23]. Pregnancy-induced hypertension was determined by hypertension without proteinuria after 20 weeks of gestation in a previously normotensive woman, and pre-eclampsia was determined by hypertension and proteinuria after 20 weeks of gestation in a previously normotensive woman [24].

Intrapartum variables collected included gestational age at delivery, type of labor (spontaneous labor, induced labor or planned cesarean delivery), mode of delivery (vaginal delivery or cesarean delivery during labor), and maternal outcome (postpartum hemorrhage (PPH)). Induction of labor was undertaken for medical and obstetric indications (GDM, intrahepatic cholestasis of pregnancy, pre-eclampsia, prenatal suspicion of SGA or LGA, and prolonged pregnancy). Gestational age at delivery was determined by the craniocaudal length at a first-trimester ultrasound examination or by the date of the last menstrual period and/or a second- or third-trimester ultrasound if the first-trimester ultrasound was not performed) [25]. Pre-term delivery was defined as gestational age at delivery < 37 weeks. Prolonged pregnancy was determined by gestational age ≥ 41(+0) weeks [25]. PPH was defined by a blood loss ≥ 500 mL, assessed with a collector bag placed just after birth.

The neonatal outcome collected included birth weight, fetal sex, birth weight z-score (BWz), cephalic perimeter, umbilical artery blood gas values that were routinely measured, and immediate neonatal data that were recorded with a systematic pediatrician examination after delivery: 5 min Apgar score, need for resuscitation or intubation, admission to the neonatal intensive care unit (NICU), and neonatal death. BWz were calculated according to French published references, adjusted for sex and term at birth [26]. Neonatal morbidity was a composite variable, defined by at least one of the following criteria: Preterm delivery (<37 weeks), 5 min Apgar score less than 7, pH less than 7.10, need for resuscitation or intubation, NICU admission, and neonatal death.

### 2.4. Biological Tests

All the included women had a measurement of the trace metals as part of their routine follow-up fasting venous blood tests at the second trimester of pregnancy. Blood samples during pregnancy were collected at fasting in the polypropylene tubes, were immediately centrifuged, and the supernatant was then stored at −20 °C until analysis. All measures of the trace elements were done in the Department of Biology by the same biologist (VDR). Biological parameters were assessed using routine techniques. Serum zinc was measured quantitatively by atomic absorption spectrometry, and quantitative determination of serum selenium levels were measured by inductively coupled/plasma mass spectrometry. Serum vitamins A1 and C levels were measured quantitatively by high-performance liquid chromatography analysis, and serum vitamins B1 and B6 levels were measured by high performance liquid chromatography-tandem mass spectrometry.

During pregnancy, there are no specific validated standards for the required levels of vitamins and trace elements. Values for micronutrients deficiencies (defined as serum trace element level below the lower reference level) were based on usual observed values in general healthy pregnant women at the second trimester of pregnancy from a comprehensive review of the literature [27], and based on recent French clinical practice guidelines for pregnancy after BS [11].

### 2.5. Statistical Analysis

The finalized dataset of included pregnant women was entered into EXCEL spreadsheets (all patients’ identifiable data were excluded) and migrated into the Software for Statistics and Data Science (STATA, version 16.0, StataCorp LLC, College Station, TX, USA) for analyses. Continuous variables were described by their mean ± SD and compared between the groups by the Student’s t-tests (or Mann-Whitney U test when appropriate) and categorical variables were described by percentages and compared between the groups by the chi-squared test (or Fisher’s exact test when appropriate). No formal sample size was calculated as we analyzed data on all cases (RYGB and SG) over a specified time period. The maternal and neonatal outcome and the serum micronutrients values were compared according to the type of surgery. Spearman’s correlations were used to describe the associations between serum micronutrients values at the second trimester and neonatal outcome. A two-sided *p*-value < 0.05 indicated statistical significance.

## 3. Results

During the study period, 19,778 births took place in our tertiary public hospital, and 102 pregnant women with a history of BS were managed in our hospital. Ten pregnant women with a history of other types of BS (nine cases after laparoscopic gastric banding, and one case after duodenal switch), one pregnant woman without gestational age dating before 24 weeks, two women with multiple pregnancies, one woman with medical-indicated second trimester termination of pregnancy, and one woman with intra-uterine death before 22 weeks were excluded (Figure 1). Finally, the study included 87 pregnant women with a history of BS (RYGB (*n* = 37; 42.5%) or SG (*n* = 50; 57.5%)), a live singleton fetus at birth, and regular multidisciplinary evaluation in our ambulatory obstetric unit with serum dosages of the micronutrients.

The maternal characteristics and maternal and neonatal outcomes according to the type of bariatric surgery are shown in Table 1. The mean maternal age was 32.4 ± 5.0 years. The mean time to conception from bariatric surgery was 49.3 months (range 1–252 months), and the mean weight and BMI at the beginning of the pregnancy were 80.6 ± 2.7 kg (range, 61–132 kg) and 31.6 ± 6.6 kg/m^2^ (range, 18.4–53 kg/m^2^), respectively. No specific complication of BS, as internal hernia or obstruction, bowel intussusception or volvulus, and gastric or jejunal perforation, was reported in our study.

In the case of RYGB, the maternal age was significantly higher (34.3 ± 4.9 vs. 31.1 ± 4.6 years; *p* = 0.003), the pre-pregnancy BMI was significantly lower (29.8 ± 5.6 vs. 32.9 ± 7.1 kg/m^2^; *p* = 0.03), and the pregnancy timing after surgery was significantly higher (64.5 ± 47.0 vs. 38.1 ± 28.1 months; *p* = 0.002). Maternal and neonatal outcomes were not different depending on the type of BS (Table 1). Specifically, the mean birth weights (range) were similar between groups: 3227g (range, 2050–4160 g) in the RYGB group and 3151 g (range, 2330–3940 g) in the SG group (*p* = 0.42). The incidence of SGA (10.8% in the RYGB group vs. 6.0% in the SG group; *p* = 0.45) and neonatal morbidity (18.9% vs. 22.0%; *p* = 0.74) were similar between the groups (Table 1).

Micronutrients values and deficiencies according to the type of bariatric surgery are exposed in Table 2. The mean gestational age at the dosage of the micronutrients was 28.3 ± 2.9 weeks.

The zinc, selenium, vitamins A1, B1, B6, C, and E plasma levels were not different depending on the type of BS. In our study, all the included women, except one, have at least one micronutrient deficiency at the second trimester. Zinc deficiency was significantly more frequent after RYGP, compared to SG (18.9% vs. 0%; *p* = 0.02). Other rates of micronutrients deficiencies were similar between the groups. Additional nutritional supplementations after blood tests were similar in the groups (Table 2).

Selenium deficiency (<60 µg/L) was found in 15/87 women (17.2%): 21.6% after RYGB (8/37) and 14.0% after SG (7/50) (*p* = 0.36). Pre-term birth occurred in five cases (5.7%), all after SG and without selenium deficiency. The mean birth weight was significantly lower in the selenium deficiency group (3137 ± 550 vs. 3535 ± 737 g; *p* = 0.04), but selenium levels (67.0 ± 7.2 vs. 66.9 ± 12.0 µg/L; *p* = 0.93) and selenium deficiency (14.3% vs. 17.5%; *p* = 1.00) were similar in SGA, compared to eutrophic newborn infants. Nevertheless, selenium deficiency was negatively correlated with birth weight (*r* = −0.23; *p* = 0.03) and with BWz (*r* = −0.26; *p* = 0.01), but was not correlated with the birth weight (*p* = 0.52) or with BWz (*p* = 0.82) after adjustment for the type of surgery.

Zinc deficiency (<0.51 mg/L) was found in 7/87 women (8.0%), all after RYGB (7/37, 18.9% vs. 0%; *p* = 0.02), and pre-term birth occurred in 100% of these cases. The mean birth weight was similar in the zinc deficiency group, compared to the normal serum zinc level group (3400 ± 560 vs. 3170 ± 615 g; *p* = 0.67). Zinc deficiency was not correlated with birth weight and with BWz.

Other micronutrients deficiencies were not correlated with birth weight and with BWz using Spearman’s correlation test (Table 3).

## 4. Discussion

In our prospective, the observational study among women with previous BS, the main findings of our study are (1) serum micronutrient levels (zinc, selenium, vitamins A1, B1, B6, C, and E) at the second trimester of pregnancy were similar in women after RYGB or SG, (2) zinc deficiency was significantly more frequent after RYGP, and (3) selenium deficiency was negatively correlated with birthweight and with birthweight z-score, but not correlated after adjustment for the procedure.

Our results were in accordance with recent literature on micronutrients levels and deficiencies during pregnancy after SG or RYGB [13,14,15]. In the largest prospective observational study, which included 123 pregnancies (77 after RYGB and 46 after SG), women after SG had the same total number of nutritional deficiencies at the second trimester than after RYGB [15]. Other studies, with a smaller sample size, also found no difference in the types of nutritional deficiencies after RYGB compared to SG in well-substituted women during pregnancy [13,14]. However, in contrast with the literature, we reported similar maternal and neonatal outcomes in women with previous RYGB compared to SG. Some maternal characteristics were different between the groups (age, BMI, time to conception from BS, GWG) but none of them was correlated to birth weight and with BWz (Table 3). We may hypothesize that these results are associated with strict management of women after BS with strict monitoring of the levels of micronutrients during pregnancy in our center. Numerous studies have found adverse perinatal outcome after RYGB, such as increased rate of pre-term delivery and higher risk of SGA, compared to the outcome from women with a similar BMI and who had not undergone BS [28,29]. Some retrospective studies with fragmented information on micronutrients supplementations during pregnancy have also suggested that RYGB leads to a lower mean birth weight, compared to SG [30]. The micronutrients deficiencies in women after RYGB should be associated with impaired fetal growth due to a poor dietary pattern combined with gastrointestinal modifications induced by the surgery. These modifications may theoretically increase the risk of neonatal adverse events due to micronutrients deficiencies during pregnancy [31]. Nevertheless, two large systematic reviews found minimal evidence of neonatal adverse events of micronutrients deficiencies during pregnancy with previous BS, specifically due to maternal deficiencies of vitamins A, B12 and K, and folate [12,32]. A more recent systematic review including 27 studies comprising 2056 women with pregnancies after BS reported various micronutrients deficiencies (vitamins A, B12, D, and iron) during pregnancy, but evidence of adverse neonatal outcome related to these deficiencies is lacking [33]. In a recent study with very well-substituted women with a particular focus on the levels of micronutrients during pregnancy, mean birth weight, mean BWz, and incidence of SGA were similar after RYGB and SG [15]. As in our study, these results are probably due to the strict management of women with previous BS during pregnancy in an ambulatory procedure with adapted additional supplementations after blood tests during pregnancy; adverse fetal outcomes may reflect the poor nutritional status of pregnant women who have had BS. Recent guidelines currently recommend routine testing and dietary supplementation before and after bariatric surgery [10], and, specifically, during pregnancy [11].

We have also found that zinc deficiency (7/87) was significantly more frequent after RYGP, compared to SG. Zinc is a trace mineral used as a co-factor for enzymes, and similar to other divalent cations (iron, copper, calcium), zinc absorption requires an acidic environment in the stomach and is absorbed in the proximal intestine. Not surprisingly, zinc absorption was demonstrated to have decreased to 42% and 65% of the baseline at 6 and 18 months after RYGB [34]. Furthermore, narrative reviews concluded that zinc deficiency may shorten pregnancy, and specifically after RYGB [35,36]. In our study, zinc deficiency occurred in 7/87 women (8.0%), all after RYGB, and pre-term birth occurred in 100% of these cases. Zinc deficiency during pregnancy in women with previous RYGB should also induce limited fetal growth [35,36]. Nevertheless, our findings did not support these results without any increasing rate of adverse neonatal events (SGA, BWz) in cases of zinc deficiency, probably due to the strict management of women with previous BS during pregnancy in an ambulatory procedure with adapted additional supplementations after blood tests during pregnancy.

Regarding neonatal issues, the mean birthweight was significantly lower in women with selenium deficiency at the second trimester and a history of BS (RYGB or SG), and selenium deficiency was negatively correlated with birthweight and with birthweight z-score, but not correlated after adjustment for the procedure. Selenium is an antioxidant ranging from antioxidant to anti-inflammatory effects, and selenium deficiency has been associated with an increased risk of mortality, poor immune function, and cognitive decline [37]. After BS, a recent narrative review including six studies and a total of 531 patients showed that selenium deficiency has been observed between 3% and 12% within 24 months after RYGB [38]. During pregnancy, the low selenium status is associated with recurrent abortion, pre-eclampsia, and altered fetal growth [36]. However, in women with previous RYGB and selenium deficiency during pregnancy, evidence of neonatal adverse events are lacking. To date, the literature review does not contain any study on selenium deficiency and altered fetal growth in pregnant women after previous BS.

The principal strength of our study is that all included pregnant women with previous RYGB or SG in our center were managed by the same obstetric and endocrinology team in an ambulatory procedure throughout the study period, which avoided significant variation regarding clinical and biological management, and which avoided uncompleted information the additional nutritional supplementations according to blood tests. Second, we reported a strict monitoring of the levels of micronutrients and systematic additional supplementations after blood tests throughout the pregnancy. In the literature, micronutrients deficiencies during pregnancy have been associated with an increased rate of SGA [12,13], except in the case of strict monitoring and management of the levels of micronutrients [15].

Our results must be interpreted in light of certain limitations. First, our study reflects the experience of one tertiary hospital and its results can be generalized only to other institutions with bariatric expertise and similar obstetric care in women with a history of BS (systematic ambulatory procedure, systematic and repeated biological evaluation of the levels of micronutrients during pregnancy, additional nutritional supplementations according to blood tests). Second, as a recent prospective study on the perinatal issues of micronutrients deficiencies during pregnancy after BS [15], we have chosen to focus on the second trimester to evaluate the serum levels of micronutrients. We have not analyzed variations of blood levels of each micronutrient or severity of each deficiency, and that may represent other limitations of our study. However, nutritional parameters always vary during pregnancy and, specifically, among women after a previous BS with additional nutritional supplementations according to blood tests. Moreover, patients undergoing BS know they are under the risk of micronutrients deficiencies [39,40] and multivitamin supplements are routinely recommended lifelong to prevent vitamin deficiencies and long-term complications [10]. Nevertheless, despite all these precautions, recent reviews reported an inconsistent adherence of multivitamin supplements intake after BS [9,41]. Nevertheless, studies on the optimal regimens and micronutrients requirements in pregnancies after BS are lacking, large cohort studies are needed to increase the level of knowledge on the prevalence and incidence rates of micronutrients deficiencies and to document the effect of supplementation regimens during pregnancy. Third, the number of included women (*n* = 87) with a limited number of neonatal complications (pre-term delivery: *n* = 5; SGA: *n* = 7) in our sample might not have been high enough to reveal a clinically meaningful association between micronutrients deficiencies and altered fetal growth according to the type of BS and after adjustment of potential confounders.

## 5. Conclusions

In our experience, selenium deficit at the second trimester in women with a history of BS is negatively correlated with birthweight and with birthweight z-score. The zinc level was also significantly lower after SG compared to RYGB. The levels of micronutrients surveillance along pregnancy in women with a history of BS is necessary to decrease the risk of inadequate fetal growth in the patients. More research is required to inform evidence-based recommendations for improving the nutritional status during pregnancy following BS and optimizing the nutritional and neonatal outcome.

## Figures and Tables

**Figure 1 jcm-10-00204-f001:**
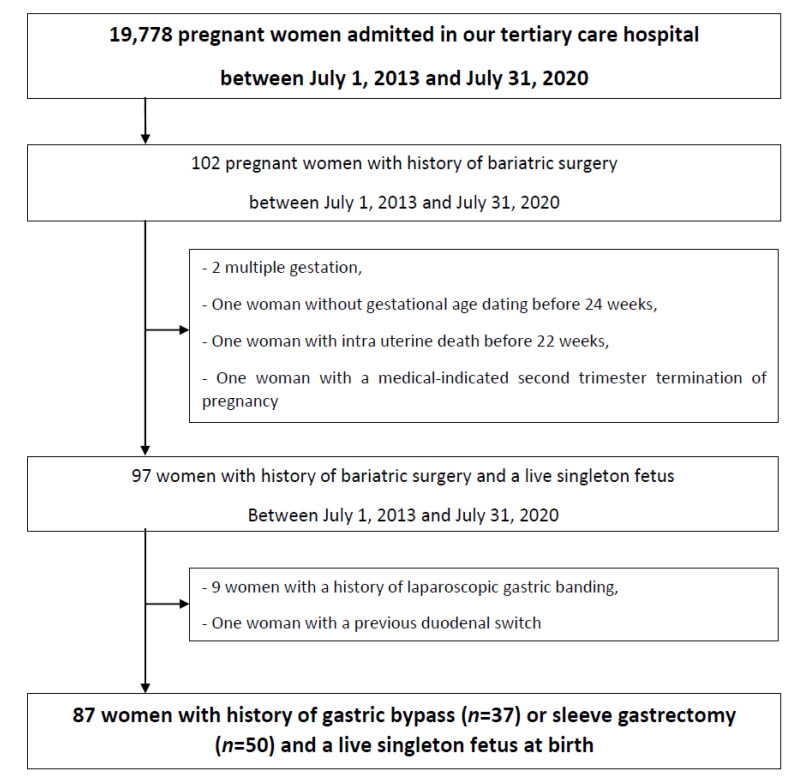
Flow chart.

**Table 1 jcm-10-00204-t001:** Maternal characteristics and maternal and neonatal outcomes according to the type of bariatric surgery.

	Roux-en-Y Gastric Bypass, *n* = 37 (42.5%)	Sleeve Gastrectomy, *n* = 50 (57.5%)	*p*-Value
Age, years	34.3 ± 4.9	31.1 ± 4.6	0.003
Caucasian	36 (97.3)	47 (94.0)	0.63
Pre-pregnancy BMI, kg/m^2^	29.8 ± 5.6	32.9 ± 7.1	0.03
Obesity (BMI ≥ 30 kg/m^2^)	16 (43.2)	32 (64.0)	0.06
Pre-existing type 1 or 2 diabetes	2 (5.4)	3 (6.0)	0.93
Chronic hypertension	1 (2.7)	1 (2.0)	0.85
Smoking during pregnancy	7 (18.9)	19 (38.0)	0.06
Time to conception from BS, months	64.5 ± 47.0	38.1 ± 28.1	0.002
Nulliparity	10 (27.0)	16 (32.0)	0.62
Gestational weight gain, kg	10.2 ± 5.4	10.3 ± 4.7	0.87
Adequate weight gain during pregnancy	25 (67.6)	30 (60.0)	0.47
GDM	11 (29.7)	13 (26.0)	0.70
Pregnancy-induced hypertension/Pre-eclampsia	2 (5.4)	5 (10.0)	0.48
Intrahepatic cholestasis of pregnancy	0	1 (2.0)	0.66
Additional supplementations after blood tests	12 (32.4)	16 (32.0)	0.96
Gestational age at delivery, weeks	39.2 ± 1.1	39.0 ± 1.8	0.51
Spontaneous labor	19 (51.4)	23 (46.0)	0.63
Induced labor	16 (43.2)	16 (32.0)	0.29
Mode of delivery			
Planned cesarean delivery before labor	2 (5.4)	11 (22.0)	0.03
Cesarean delivery during labor	3 (8.1)	5 (10.0)	0.79
Vaginal delivery	32 (86.5)	34 (68.0)	0.05
Postpartum hemorrhage	3 (8.1)	4 (8.0)	1.00
Pre-term delivery (<37 wk)	0	5 (10.0)	0.07
Birth weight, g	3227 ± 499	3151 ± 373	0.42
Birth weight z-score	0.2 ± 1.1	0.2 ± 0.9	0.83
SGA	4 (10.8)	3 (6.0)	0.45
Birth weight < 2500 g	3 (8.1)	1 (2.0)	0.23
LGA	3 (8.1)	2 (4.0)	0.65
Birth weight > 4000 g	1 (2.7)	0	0.66
Cephalic perimeter, cm	33.0 ± 1.7	33.0 ± 1.5	0.83
5 min Apgar score < 7	3 (8.1)	3 (6.0)	0.70
pH < 7.10	0	3 (6.0)	0.26
Need for resuscitation or intubation	0	0	-
NICU admission	4 (10.8)	10 (20.0)	0.25
Neonatal death	0	0	-
Neonatal morbidity	7 (18.9)	11 (22.0)	0.74

Values are given as the mean ± SD or number (percentage) unless otherwise indicated. BS: Bariatric surgery; BMI: Body mass index; GDM: Gestational diabetes mellitus; SGA: Small-for-gestational age; LGA: Large-for-gestational age; NICU Neonatal intensive care unit.

**Table 2 jcm-10-00204-t002:** Micronutrients values and deficiencies according to the type of bariatric surgery.

	Roux-en-Y Gastric Bypass, *n* = 37 (42.5%)	Sleeve Gastrectomy, *n* = 50 (57.5%)	*p*-Value
Gestational age at blood samples, weeks	27.9 ± 3.2	28.7 ± 2.6	0.18
Additional supplementations after blood tests	12 (32.4)	16 (32.0)	0.96
Se (µg/L)	66.3 ± 13.5	67.4 ± 10.2	0.68
Deficiency *	8 (21.6)	7 (14.0)	0.36
Zinc (mg/L)	1.0 ± 2.5	1.0 ± 2.0	0.93
Deficiency *	7 (18.9)	0	0.002
Vitamin A1 (µg/L)	496.4 ± 623.6	503.3 ± 171.0	0.94
Deficiency *	13 (35.1)	12 (24.0)	0.26
Vitamin B1 (µg/L)	54.6 ± 18.2	55.4 ± 15.2	0.83
Deficiency *	4 (10.8)	2 (4.0)	0.40
Vitamin B6 (µg/L)	5.1 ± 6.5	3.8 ± 3.5	0.24
Deficiency *	29 (78.4)	44 (88.0)	0.23
Vitamin C (mg/L)	4.1 ± 3.0	4.8 ± 3.4	0.32
Deficiency *	30 (81.1)	33 (66.0)	0.12
Vitamin E (mg/L)	12.4 ± 3.6	13.2 ± 4.0	0.11
Deficiency *	8 (21.6)	4 (8.0)	0.07

Values are given as the mean ± SD or number (percentage) unless otherwise indicated. * Deficiency was defined according to normal reference values in pregnant women at the second trimester of pregnancy [11,27].

**Table 3 jcm-10-00204-t003:** Characteristics that were correlated with birth weight and with birth weight z-score using Spearman’s correlation test.

Variables	Birth Weight		Birth Weight Z-Score	
	Spearman’s Correlation Coefficient	*p*-Value	Spearman’s Correlation Coefficient	*p*-Value
Age, years	0.010	0.93	0.012	0.91
BMI, kg/m^2^	0.188	0.08	0.191	0.11
Time to conception from BS, months	0.126	0.25	0.047	0.67
Gestational weight gain, kg	0.100	0.36	−0.087	0.42
Selenium deficiency	−0.230	0.03	−0.265	0.01
Zinc deficiency	−0.189	0.08	−0.165	0.13
Vitamin A deficiency	−0.034	0.76	−0.048	0.67
Vitamin B1 deficiency	−0.246	0.06	−0.238	0.06
Vitamin B6 deficiency	−0.055	0.61	−0.033	0.76
Vitamin C deficiency	−0.088	0.42	−0.224	0.06
Vitamin E deficiency	−0.094	0.39	−0.128	0.42
Gestational age at birth, weeks	0.424	<0.001	−0.281	0.008

BMI: Body mass index.

## Data Availability

The data presented in this study are available on request from the corresponding author. The data are not publicly available due to institutional policy.

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
