# Peer review of "A Prospective Study of Association of Micronutrients Deficiencies during Pregnancy and Neonatal Outcome among Women after Bariatric Surgery"

_jcm, 2021, doi:10.3390/jcm10020204_

Round 1
Reviewer 1 Report
The study aimed to evaluate serum levels of micronutrients during pregnancy among women after bariatric surgery (RYGB or SG) and to assess the association between micronutrient deficiencies and neonatal outcome.
This is an interesting paper, yet I have are several comments:
There is no comment or explanation in the discussion regarding the differences in maternal characteristics between the two types of surgeries that may impact the results. For example, does the difference in BMI or obesity between RYGB and SG impact the results? Or does the significant difference in time from BS to conception effect the micronutrient deficiencies?
As for the results regarding neonatal outcomes:
The comparison between women with micronutrient deficiencies and the control group of eutrophic newborn infants and neonatal outcomes should be presented in a table or graph.
There is no comment about the level of deficiency and whether it has an impact on the results.
The authors only mention the mean birth weight. The range of birth weights should be reported as well.
The sentence in lines 251-254 is too long and unclear. It should be divided into two sentences.
The limitation about the patients' adherence to intake of multivitamin supplements after BS is important, yet the sentence regarding this issue is long and unclear (lines 308-312). Again, it should be divided into two sentences.
Reviewer 2 Report
The manuscript by Ducarme et al. addresses an interesting and relevant subject and is warranted in the discussion regarding the effects of gastric bypass surgery on pregnancy outcomes. I however have some comments:
- Why were the levels of micronutrients only evaluated at second trimester?
- It would have been interesting to have information on complications in pregnancy due to the bariatric surgery. Are there any information on that?
- It is not completely clear how the GDM diagnosis was made in the BS women: was it only by a 1-week selfmonitoring of capillary blood glucose?
- The authors perform Spearman's correlations to describe the associations between serum micronutrients values and neonatal outcomes. It is recommendable that these values were added to a table 3.
- The levels of B12, Vitamin D, calcium and iron are also most relevant in patients who have had BS and are especially important in pregnant women with previous BS. Howcome these parameters are not included?
- Line 273-278: In the discussion section about zinc deficiency it should be mentioned that zinc deficiency was found in women with RYGB and that preterm delivery occurred in all cases (100 %) as mentioned in line 230-233, as this is a very important finding regarding zinc deficiency, if I understand it correctly.
- Line 43: "Since" should be changed to "within the last 10 years"
Reviewer 3 Report
It is a good clinical research that reflects the experience of a tertiary level center of perinatal medicine and, as you describe in the test, the results may be valid for other populations of patients. However it adds new data on this subject.
Round 2
Reviewer 2 Report
The authors have addressed my points of criticism satisfactory.
